# Bio-Doped Microbial Nanosilica as Optosensing Biomaterial for Visual Quantitation of Nitrite in Cured Meats

**DOI:** 10.3390/bios12060388

**Published:** 2022-06-03

**Authors:** Siti Nur Syazni Mohd Zuki, Choo Ta Goh, Mohammad B. Kassim, Ling Ling Tan

**Affiliations:** 1Southeast Asia Disaster Prevention Research Initiative (SEADPRI), Institute for Environment and Development (LESTARI), Universiti Kebangsaan Malaysia, Bangi 43600, Selangor Darul Ehsan, Malaysia; p94593@siswa.ukm.edu.my (S.N.S.M.Z.); gohchoota@ukm.edu.my (C.T.G.); 2Department of Chemical Sciences, Faculty of Science and Technology, Universiti Kebangsaan Malaysia, Bangi 43600, Selangor Darul Ehsan, Malaysia; mb_kassim@ukm.edu.my

**Keywords:** biosensor, nitrite, silaffin, silica, optic

## Abstract

A microbial optosensor for nitrite was constructed based on biomimetic silica nanoparticles, which were doped with R5, a polypeptide component of silaffin, as a robust biosilica immobilization matrix entrapped with *Raoultella planticola* and NAD(P)H cofactor during the in vitro biosilicification process of silica nanoparticles. Ruthenium(II)(bipy)2(phenanthroline-benzoylthiourea), the chromophoric pH probe, was physically adsorbed on the resulting biogenic nanosilica. Optical quantitation of the nitrite concentration was performed via reflectance transduction of the bio-doped microbial nanosilica at a maximum reflectance of 608 nm, due to the deprotonation of phen-BT ligands in the ruthenium complex, while the intracellular enzyme expression system catalyzed the enzymatic reduction of nitrite. Reflectance enhancement of the microbial optosensor was linearly proportional to the nitrite concentration from 1–100 mg L^−1^, with a 0.25 mg L^−1^ limit of detection and a rapid response time of 4 min. The proposed microbial optosensor showed good stability of >2 weeks, great repeatability for 5 repetitive assays (relative standard deviation, (RSD) = 0.2–1.4%), high reproducibility (RSD = 2.5%), and a negligible response to common interferents found in processed meats, such as NO_3_^−^, NH_4_^+^, K^+^, Ca^2+^, and Mg^2+^ ions, was observed. The microbial biosensor demonstrated an excellent capacity to provide an accurate estimation of nitrite in several cured meat samples via validation using a standard UV-vis spectrophotometric Griess assay.

## 1. Introduction

The immobilization of biological molecules with catalytic and recognition functions within robust polymer matrices remains a challenging task, despite considerable research being focused on this area [1]. Efficient bio-immobilization depends on the perfect matching of several factors, e.g., the proteins, the biorecognition process, the supporting matrix, and the additives used for support modification. The selection of an appropriate substrate material can literally influence the enzyme physical-chemical properties in terms of diffusion and catalytic efficiency [2]. 

Enzyme immobilization by sol-gel entrapment has been used for many decades due to the ease of surface functionalization, high surface area, mechanical and thermal stability, and resistance to both chemical and biological agents [3]. Silicates, mainly in the form of composite or functionalized mesoporous silica can be acquired by two principal methods, i.e., the Stöber process and the microemulsion method, employing ammonia catalyzed hydrolysis and condensation of ethoxysilanes in alcohol to produce the uniform Stöber silica particles, with potential application in pharmaceuticals via enzyme encapsulation, drug delivery, and cell markers, as well as in other industries via electronic device catalysis, insulators, etc. [4]. In view of the harsh conditions required for chemical synthesis associated with the conventional sol-gel methods, there is a limit for enzyme loading capacity in the silica gels, primarily due to the hydrophobicity of the alkyl silicates, denaturing effect from alcohol byproduct, and aggregation of protein molecules at elevated concentrations. These have tremendously restricted their applicability in the field of applied biotechnology, biocatalysts, biosensors, biodiagnostics, and biomedicine [1,5].

Sol-gel bioencapsulation of enzymes based on poly(glyceryl silicate) has a higher biocompatibility, which enables the efficient confinement of biological moieties, e.g., proteins and cells inside the silica gel with high load capacity. However, the differences in the precursor hydrolysis or condensation process, precursor toxicity, and pore structure may contribute to the efficiency of the poly(glyceryl silicate) as an enzyme immobilization support matrix [1]. A high enzyme recovery yield has been reported for silica xerogel treated with protic ionic liquid and bifunctional agents, such as glutaraldehyde and epichlorohydrin, as a novel support strategy for the effective immobilization of enzyme lipase by covalent binding. Using a treated silica xerogel for support exhibited a better enzyme activity performance compared to those sol-biogels fabricated from tetramethoxysilane, poly(methyl silicate), and glycerol-doped poly(silicic acid) [2]. The biomimetic synthesis of silica has attracted much attention recently due to its mild synthesis conditions and facile synthesis strategies. Silaffin (R5) is a polycationic peptide involved in the formation of the intricate silica cell wall of the diatoms of *Cylindrotheca fusiformis* (*C. fusiformis*). Diatoms are the largest group of unicellular eukaryotic microalgae and are the dominant phytoplankton in the oceans [6,7]. Silaffin can act as an organic matrix to induce the in vitro silica genesis, forming nano-structured silica precipitates containing many pores [7,8]. The entrapment of biological recognition molecules by biogenic silica production creates a favorable setting for accessing diffused analyte molecules [8].

Nitrite is primarily used in food processing to preserve meat-based products, such as ham, sausage, bacon, etc., in order to prevent the risk of bacterial contamination, since nitrite imparts antioxidant and antimicrobial properties to meat and meat products [9]. Moreover, the addition of nitrite to meat products can also impart a better flavor, taste, and aroma, and to cure the red-pinkish color of meats so as to extend their shelf life [10]. Regrettably, nitrite can react with amines or amides in the meat during processing, storage, and cooking to form carcinogenic N-nitroso compounds (NOCs) (i.e., N-nitrosamines and N-nitrosamides), resulting in the development of esophageal and stomach cancers with long-term consumption [10,11]. As nitrite prevents the growth and toxin production of *Clostridium botulinum*, it is permitted to be used as food additives in a wide range of foods. However, the International Agency for Research on Cancer (IARC) has recently raised concern over the adverse health effects of nitrite metabolic compounds, e.g., nitric oxide and NOCs, that favor the endogenous nitrosation of the amino acid substrate that is carcinogenic to humans [12]. The nitrite content in the processed meats was found to be in the range of 0.29 mg L^−1^ to 44.72 mg L^−1^ [13]. The Food Safety and Inspection Service (FSIS), a public health regulatory agency of the United States Department of Agriculture (USDA), allows the safe use of nitrite not to exceed 156 mg L^−1^ per 100 pounds of meat in sausage products [14].

The conventional methods of detection and determination of nitrite include gas chromatography–mass spectrometry (GC–MS), liquid chromatography with fluorescence detection (LC-FL), and ion-exchange liquid chromatography, while spectrophotometric detection methods have been proposed for the determination of nitrite in biological, food, and environmental samples [15,16,17,18]; high performance liquid chromatography with UV absorption (HPLC/DAD) using a phosphatidylcholine column has been applied for the separation and quantitative determination of nitrites in human saliva, whereby nitrite was detected after oxidation to nitrate by potassium permanganate in acidic conditions [19]; vortex-assisted supramolecular solvent-based liquid phase microextraction via the UV-vis spectrophotometric method [20] and ion chromatography (IC) coupled with conductivity detection were exploited to determine the content of nitrite ions in homogenized meat samples of baby foods [16]. However, these methods involved tedious sample pre-treatment steps, e.g., the dissolution of the sample, derivatization of aqueous nitrite via Griess chemistry, reduction of nitrate to nitrite, and extraction with toluene. Therefore, the search for a simple, fast, and more effective method is necessary to provide a way for tracking the legal recommended limits of nitrite in foods, especially cured meat products, which are often considered a “pleasure” product for people all over the world.

The ion-selective electrodes (ISEs) are part of a group of relatively simple and inexpensive analytical tools commonly referred to as sensors. The nitrite ISE electrode is intended for measuring nitrate ion concentrations and activities in aqueous solutions. Its preferred applications are to be found in the field of water, fertilizers, soil, meat, baby food analysis, etc. Commercially available ISEs for nitrite analysis that provide a wider measuring range are those such as the Thomas brand nitrite ISE (0.5–460 mg L^−1^) [21]; NTsensors CNT_ISE (0.5–1000 mg L^−1^) [22], TRUEscience ISE (0.5–460 mg L^−1^) [23]; Orion nitrite electrode (0.5–200 mg L^−1^) [24], and CLEAN instruments nitrite ISE sensor (0.2–220 mg L^−1^) [25]. These simple ISE testers can be used as alternatives to analyze meat for the presence of nitrite.

In this study, we report the first application of the silica-based cell matrix of R5 as an optical biosensing material with a reflectometric transducer, modelled using whole cell nitrite-degrading bacteria. The microbial optosensor, based on biomimetic silaffin-fusion silica nanoparticles, was fabricated with a facile route by employing silica-condensing synthetic R5 peptide (H2N-SSKKSGSYSGSKGSKRRIL-COOH), the repeating unit of the silaffin polypeptide in the biosilicification reaction mixture containing hydrolyzed tetramethyl orthosilicate (TMOS), in order to physically entrap the *Raoultella planticola* (*R. planticola*) microbial cells and NAD(P)H cofactors within the sol-biogel composite during the precipitation process of biosilica nanoparticles. The biomimetic approach to biological immobilization via an affinity mechanism, i.e., by using the silaffin peptide sequences with an affinity for silica material, promotes biocompatibility without any formation of covalent bonds to the receptor molecules. This eliminates the major problem with the covalent bonding method which is that the chemical modification of the biological molecule could result in the loss of the functional conformation of the biological molecules. The immobilized ruthenium bipyridine complex pH optical label at the biomimetic nanosilica surface enables visual detection of nitrite by a simple color change of the bio-doped whole cell bacterial biosensor from orange to yellow.

## 2. Materials and Methods

### 2.1. Chemicals

Sodium nitrite was purchased from Systerm and was dissolved in 0.1 M potassium phosphate (K-phosphate) buffer at pH 7.4. Nicotinamide adenine dinucleotide phosphate sodium (NAD(P)H, Roche Diagnostic GmbHA) was prepared in 0.1 M K-phosphate buffer (pH 7.4) at 40 mg L^−1^. Synthetic peptide sequence: H_2_N-SSKKSGSYSGSKGSKRRIL-COOH with 95% purity of HPLC was synthesized (purchased from Apical Scientific Pvt. Ltd.) The stock solution of R5 peptide was prepared in ultrapure water at 100 mg mL^−1^. TMOS and hydrochloric acid (HCl) were each obtained from Merck. All the chemicals used in this work were of analytical grade and were used as received without further purification. Ruthenium complex, [Ru(II)(bpy)_2_(phen-BT)](PF_2_)_6_, which has been synthesized in the previous study by Tan et al. [26], and *R. planticola* bacteria cells, which have been isolated from local edible bird’s nests from the previous study [27], were utilized as the model microbial intercellular enzymes in the present study. The *R. planticola* bacteria cells were stored at −80 °C when not in use. They were allowed to thaw on ice prior to mixing with 2 mM NAD(P)H at a volume ratio of 3:1 for the subsequent fabrication of the microbial optosensor.

### 2.2. Instrumentations

A Varian Cary 50 UV-vis spectrophotometer was used for absorption measurement of the microbial intracellular enzymatic reaction in the solution with the ruthenium complex optical pH probe. Reflectance measurement of the optosensing biomaterial was performed by using a fibre optic reflectance spectrophotometer (Ocean Optic SD2000) with a DH-2000-BAL UV-Vis-NIR light source within the wavelength range of 200–1099 nm. Ultrapure water with a conductivity of 0.055 µS cm^−1^ (18.2 MΩ·cm) obtained from the Sartorius Ultrapure Water Purification System, model arium^®^ pro DI (Sartorius, Göttingen, Germany), was used to prepare all the chemical and biological solutions. The size and morphology of the microbial biomimetic nanosilica were determined using a field emission scanning electron microscope (FESEM, brand Zeiss, model Merlin) at an accelerating voltage of 3 kV (STEM mode) and magnification of 50 k×.

### 2.3. Biomimetic Synthesis of Silica nanoparticles and Fabrication of Microbial Optosensor

The biomimetic nanosilica was synthesized according to Luckarift et al. [5], with some modifications. Silicic acid was prepared by mixing TMOS with 1 mM of HCl to a final concentration of 1 M. The biomimetic silicification reaction consisted of 10 μL of 10 mg mL^−1^ hydrolyzed TMOS, 10 μL of 100 mg mL^−1^ R5 peptide solution, and 80 μL of mixed *R. planticola*/2 mM NAD(P)H suspension in a volume ratio of 3:1. The mixture was agitated at room temperature (25 °C) to allow the silaffin peptide to catalyze the formation of silica nanoparticles in vitro at ambient conditions. The as-synthesized microbial R5-fusion nanosilica was then washed with an abundance of ultrapure water, and reconstituted in 0.1 mg mL^−1^ orangish ruthenium [Ru(II)] complex solution at neutral pH, followed by stirring using a IKA magnetic hotplate stirrer (C-MAG HS7). The resultant orange-colored microbial silica particles were washed with a plentiful amount of ultrapure water to remove the loosely bound Ru(II) complexes. A total of 100 mg of the microbial biomimetic nanosilica was deposited in a round plastic case at 4 mm diameter and kept in a refrigerator at 4 °C. The as-prepared ruthenium-modified microbial optosensor was characterized biochemically for optimum pH, linear detection range, response time, operational stability, repeatability, reproducibility, and interference effect. 

### 2.4. Validation Study

The microbial optosensor based on biomimetic silica nanoparticles was validated with the Griess test as the standard method for the analysis of nitrite, as it is widely used by regulatory agencies as an official method for nitrite quantification in water and food samples. Several cured meat samples, such as nuggets, sausage, and canned meats, were prepared and pre-treated according to the manual of methods for food analysis of the Food Safety and Standards Authority of India [28], which provides clear-cut instructions as to how processed meat samples are prepared before analysis. Approximately 5 g of each processed meat sample was weighed out, chopped to mince, and transferred into a 250 mL volumetric flask containing 50 mL of hot water at 80 °C to homogenize. The heating process was proceeded in a steam bath for 2 h with continuous shaking. The resulting liquid food samples were cooled to room temperature, centrifuged to collect the supernatant clear solutions, and kept at 4 °C until use. A standard calibration curve for nitrite was established by using Griess reagent containing *N*-1-naphthylethylenediamine dihydrochloride (NED) and sulphanilamide in 5% phosphoric acid. The optical absorbance band reading at 540 nm was recorded after leaving the mixture of Griess reagent with nitrite for 5 min at ambient conditions. The same pre-treated liquid food samples were then examined with the reflectance microbial optosensor at 608 nm under optimal conditions for nitrite determination. The statistical paired sample *t*-test technique was applied to determine significant differences between the nitrite concentrations obtained by both the UV-vis spectrophotometric Griess assay and the reflectance microbial biosensor. 

## 3. Results and Discussion

### 3.1. Morphology of R5-Fusion Microbial Nanosilica and Optical Nitrite Biosensing Response

The biomimetic entrapment method is an effective and sturdy immobilization technique to hold biologicals or active substances of interest onto silica particles by mimicking the macromolecules in the silicification process [29,30]. The excellent biomimetic support properties of the silica matrix, with a benign immobilization method, could retain more than 90% of the enzyme activity in the biosilica particles, with a negligible loss of enzyme activity during immobilization [5]. In the present study, the biosilica matrix has been synthesized via a facile biomimetic silicification process in order to physically entrap *R. planticola* microbial cells and non-protein NAD(P)H chemical compounds during the biogenic nanosilica formation. The field emission scanning electron micrograph in Figure 1 reveals that the resulting silaffin-fusion microbial silica is a network of silica nanoparticles with an irregular shape and a diameter of 50–100 nm. Physical adsorption of the ruthenium pH probe molecules at 0.1 mg mL^−1^ on the microbial nanosilica produced a homogeneous orangish silica-based cell matrix of R5, as demonstrated in the photographic image in the inset of Figure 1. 

The R5-fusion biosilica matrix was synthesized by using TMOS as a silica source and the biosilica matrix, composed of four-dimensional silicon dioxide units with high porosity. Therefore, it is postulated that the ruthenium pH probe molecules were physically bound on the biomimetic nanosilica via hydrogen bonding, since a silica-based cell matrix is a suitable adsorbent for water, alcohol, phenols, amines, etc., through hydrogen bonds at high loading capacity [31,32]. The intense orangish biosilica of R5 would allow a noticeable color change of the microbial optosensor ascribed to deprotonation of phen-BT ligands in the ruthenium complex, while the intracellular enzyme expression system catalyzed the enzymatic reduction of nitrite, resulting in the yellowish substrate, which can be visually observed. 

A high concentration of Ru(II) complex in solution i.e., 0.1 mg mL^−1^ exhibits a very intense orange color because it absorbs light in the blue-green region of the visible light spectrum, and orange is the complementary color of blue-green. At a lower concentration of 0.01 mg mL^−1^, the Ru(II) complex solution turns a yellow hue, with a maximum UV-vis absorbance wavelength recorded at 455 nm (Figure 2a). The addition of whole cell *R. planticola* and cofactor NAD(P)H, however, slightly diluted the yellowish 0.01 mg mL^−1^ Ru(II) complex solution, rendering a slight decrease in the absorption peak response at 455 nm (Figure 2b). Further addition of nitrite into the Ru(II) complex solution in the presence of the microbial intracellular NAD(P)H nitrite reductase (NNR) enzyme and NAD(P)H co-substrate was observed to discolor the solution, causing the absorption band at 455 nm to diminish (Figure 2c). This implies that the nitrite-degrading bacteria has completely catalyzed the conversion of nitrite into ammonium hydroxide (NH_4_OH), making the solution more basic and leading to a dramatic increase in the pH, inducing the deprotonation of the phenanthroline-benzoylthiourea (phen-BT) molecules in the ruthenium complex. This finding is inconsistent with the previous study carried out by Tan et al. [26] related to pH sensitivity of the Ru(II) complex, whereby the Ru(II) complex in solution exhibited a strong metal-to-ligand charge transfer (MLCT) transition band at 455 nm that decreased with an increase in the solution pH from pH 1 to pH 11. As a result of these observations on pH sensitivity of the ruthenium(II)(bipy)_2_(phenanthroline-benzoylthiourea) [Ru(II)(bpy)2(phen-BT)] complex in the solution, the Ru(II)(bpy)_2_(phen-BT) complex was later employed as a pH-sensitive chromophoric probe in the solid-state optical microbial biosensing of nitrite concentration.

The R5-fusion microbial nanosilica, which was synthesized based on the biomimetic silicification process in the silicic acid solution involving R5 synthetic peptide as the biocatalyst to initiate the precipitation of silica nanoparticles with the co-silicification of microbial intracellular NNR enzyme and cofactor NAD(P)H, followed by physically coating with ruthenium complex pH optical label on the particles’ surface, has been adopted as the optical biosensor for nitrite detection. As a model system, the microbial intracellular NNR enzyme catalyzes the reduction of the NO_2_^−^ ion to ammonium hydroxide (NH_4_OH), and concomitantly oxidizes the cofactor NAD(P)H to NADP^+^ (Equation (1)). The immobilized orangish ruthenium bipyridine complex pH indicator dye at the biosilica nanoparticles’ surface instantaneously undergoes a deprotonation reaction and changes the bio-doped whole cell bacterial biosensor to a yellowish color (Equation (2)), which can be quantified by using a reflectance spectrophotometer, as illustrated in Figure 3.
(1)$$\begin{array}{c} \text{NNR}\\ {{\text{NO}}_{2}}^{-}+3\text{NAD}(P)H+3H^{+}⇌{\text{NH}}_{4}\text{OH}+3\text{NAD}{\left(P\right)}^{+}+H_{2}O \end{array}$$
Ru(II)BTH^+^ ⇌ Ru(II)BT + H^+^(2)
(orange)      (yellow)

The distinctive color change of the microbial optosensor from orange to yellow in the presence of nitrite can be monitored by the optical reflectometric method for the quantification of nitrite concentration. The orangish microbial nanosilica presented a well-defined reflectance band, with a maximum wavelength registered at 608 nm, and the reflectance peak signal significantly heightened after enzymatic reaction with 100 mg L^−1^ nitrite, as a result of the generation of the hydroxide (OH^−^) ion from the dissociation of NH_4_OH that raised the microenvironment pH, and turned the microbial nanosilica yellow; this reflected a higher amount of light compared to its initial orange substance (Figure 4). Moreover, a leaching study of the ruthenium complex-modified bio-doped microbial nanosilica was also carried out to assess the potential release of the Ru(II) complex into 0.1 M K-phosphate buffer (pH 7.4) over a 30 min experimental period. It is imperative to note that the Ru(II) complex-modified microbial optosensor did not exhibit any leaching throughout 30 min of immersion in 0.1 M K-phosphate buffer at pH 7.4, while a negligible absorption response at 455 nm was acquired (Appendix A).

### 3.2. Effect of Buffer pH on the Microbial Optosensor Response

The ruthenium complex can structurally exist in both protonated and deprotonated forms due to the presence of the phenanthroline-benzoylthiourea structure (phen-BT). The amino and carbonyl moieties in the Ru(II) complex are pH-dependent; as such, the phen-BT moiety changes its structure according to the pH environment. These protonated and deprotonated moieties are positioned closer to the ruthenium metal and influence the characteristic absorption response as a function of pH [26]. The Ru(II) complex, therefore, shows great promise to be designated as a chromophoric pH sensitive indicator for subsequent optical nitrite biosensing studies. The buffer pH effect study on the microbial optosensor response depicted in Appendix A shows that the optimum pH of the immobilized nitrite-degrading microorganism towards the enzymatic reduction of 100 mg L^−1^ nitrite occurred at a neutral pH [33], with a remarkable reflectance signal attained at 608 nm following a perceptible color change of the biosilica nanoparticles from orange to yellow as the immobilized Ru(II) complex pH optical label underwent the deprotonation reaction. Acidic and basic conditions, however, did not favor the microbial intracellular NNR enzymatic reaction, as the surrounding acidic and alkaline environments of the intracellular enzyme expression system deviated the net charge of the molecule, which became more positively or negatively charged due to the gain or loss, respectively, of protons (H^+^); therefore, the environment could not promote further deprotonation of the Ru(II) complex at the nanosilica surface, and no noticeable color change of the microbial optosensor was perceived at a pH between pH 1.0–5.0 and pH 9.0–11.00. 

### 3.3. Response Time and Dynamic Linear Response Range of the Microbial Optosensor for Nitrite Detection

The response time of the microbial optosensor, based on biomimetic silaffin-fusion silica particles, was determined by using 400 mg L^−1^ nitrite, whereby the reflectance intensity at 608 nm was taken every 60 s for 7 min. The deprotonation reflectance band of the microbial biosilica matrix was observed to gradually increase with an increase in enzymatic reaction duration between 1 min and 5 min at pH 7.0 (Appendix A), and the optical reflectance response stabilized between 6 min and 7 min. At this point, the immobilized microbial intracellular NNR enzyme molecules within the biosilica nanospheres were considered close to saturation with the substrate and demonstrated its maximal velocity [34]. In view of the fact that the microbial biosensor was exhibiting sufficiently high optical reflectance response at 4 min of reaction time, 4 min was chosen as the optimal incubation time for the optosensing biomaterial with its analyte nitrite. The fast-responding reflectance microbial biosensor, on the order of 4 min, may be attributed to the favorable biogenic nanosilica particles that allowed rapid diffusion of analyte molecules within the substrate to access the catalytic site.

The optical evaluation of nitrite by means of reflectance transduction of the bio-doped whole cell bacterial biosensor demonstrated a broad response curve across the nitrite concentration range of 1–1000 mg L^−1^. The microbial optosensor response curve follows the Michaelis–Menten behavior, with a linear response at low analyte concentrations and a saturation response at high analyte concentrations. The apparent Michaelis–Menten constant (K_m_) of the nitrite biosensor estimated from the Lineweaver–Burk plot was 1.667 mM. Figure 5 portrays the nitrite concentration-dependent reflectometric microbial biosensor response. The increasing nitrite concentration increased the microbial intracellular NNR enzymatic reaction rate, and the microbial R5-fusion nanosilica gradually developed into a yellow hue as the immobilized ruthenium pH indicator complex underwent an increasing deprotonation reaction rate at the silaffin-fusion nanosilica surface. This greatly heightened the reflected light intensity from the resulting biomimetic silica support. The inset in Figure 5 depicts a wide dynamic linear response range, covering the nitrite concentration between 1 mg L^−1^ and 100 mg L^−1^ (R^2^ = 0.9808) of the microbial optosensor. The limit of detection (LOD) of the optical biosensor, calculated based on the average of the blank signal plus three times the standard deviation of the blank [35], was obtained at 0.25 mg L^−1^ nitrite. The dynamic linear response range study of the microbial optosensor was carried out in triplicate with three microbial biosensors (*n* = 3) prepared under the same conditions. The average reproducibility relative standard deviation (RSD) of each calibration point of the optical microbial biosensing of nitrite performed using new biosensor for each nitrite concentration testing was calculated at 2.5%. This indicates a relatively high reproducibility of the microbial optosensor, being batch-fabricated using the biosilicification entrapment process.

The large linear detection range was yielded by high bacteria loading within the matrix of fused silica nanoparticles produced by the biomimetic silicification method, and the immobilized microbial cells were more protected from the environment. The extended linear detection range of the optical microbial biosensor based on color change of the biosilica nanoparticles holds high promise for nitrite monitoring in foods, as it is often used as a food additive to stop the growth of bacteria, and to enhance the flavor and color of several meat, fish, and cheese products [36,37]. In addition, the sensor can be used for water quality monitoring, detecting nitrite as a key substance for salt waters. The overloading of nitrogen nutrients from the run-off water of agricultural land in seas, lakes, rivers, and streams can result in a series of adverse effects, known as eutrophication, that leads to massive harmful algae blooms, changes in the aquatic vegetation due to an unbalanced ecosystem, and the changed chemical composition of the water body.

### 3.4. Operational Stability and Repeatability of the Bio-Doped Microbial Biosensor

The stability of the ruthenium-modified microbial R5-fusion nanosilica has been evaluated over a 40-day period for the reflectometric determination of 20 mg L^−1^ nitrite in 0.1 M K-phosphate buffer (pH 7.0). The microbial optosensor showed a steady state reflectance response, with 100% of its initial optical reflectance intensity at 608 nm, maintained for more than 2 weeks before its optical response slowly declined for the next 4 weeks (Appendix A). The fact that the optical microbial biosensor showed good stability for its ability to maintain its initial activity for >2 weeks of investigation could be ascribed to the biosilification synthetic route to the silica nanoparticles that afforded a benign immobilization microenvironment to retain enzyme activity while permitting the access of analyte molecules to the entrapped microbial cells therein, without a diffusion barrier [7,8]. A slight diminishing of the biosensor’s reflectance response that was observed on the 20th day could probably be due to microbial degradation, and this reduced the catalytic efficiency of the immobilized nitrite-degrading microorganism towards the enzymatic reduction of nitrite, thereby interfering with the protonation reaction at the phen-BT moieties of the immobilized chromophoric pH sensitive Ru(II) complex on the surface of the R5-fusion nanosilica. The optical biosensor response continued to show deterioration from day 25 onward, which might be attributed to the irreversible loss of intracellular NNR enzyme activity following a prolonged storage period. 

Conventional sol-gel procedures involve extreme pH conditions and high alcohol concentrations, which are destructive to the mechanical stability of the resulting enzymatic biosensor [38], often leading to leakage of the receptor molecules from the sol–gel-derived silica materials, and thus, the development of a short-term-use biosensor [39]. For instance, a continuous decrease in enzymatic biosensor response can be perceived for an amperometric glucose biosensor based on silica sol-gel film due to enzyme leaching through the sol-gel matrix over the course of the biosensor stability study for a period of one month [40]. The immobilization of microorganisms in the sol-gel matrix e.g., *Escherichia coli*, *Pseudomonas*, *Streptococcus*, and *Bacillus* during the formation of silica sol-gel materials often resulted in mass mortality of bacterial cells due to the release of lower alcohols [41]. The present study revealed that the biomimetic silicification process involving R5 peptide as the additive used for nanosilica support modification has tremendously improved the mechanical stability of the resulting microbial nitrite biosensor via enhancing the long-term stability of the cells. 

The high efficacy of the biosilicification reaction immobilization technique can also be demonstrated in the repeatability study of the microbial biosensor. The microbial optosensor was repeatedly reacted with 20 mg L^−1^ nitrite after washing the bacterial cells biosensor with 0.1 M K-phosphate buffer at pH 7.0. The reflectance response at 608 nm was measured after the catalytic reaction between the immobilized biomolecules and nitrite had taken place for 4 min. Figure 6 represents the repeatability performance of the microbial silaffin-fusion nanosilica, with immobilized ruthenium bipyridine complex at the surface that functioned as the optical pH indicator. The optical microbial biosensor presented a stable deprotonation reflectance response at high reflectance intensity at 608 nm for 5 repetitive assays of 20 mg L^−1^ nitrite, with a repeatability RSD calculated in the range of 0.2–1.4%. The good repeatability performance of the microbial optosensor can be partly attributed to the mild conditions of the biosilicification immobilization method, which minimized intracellular enzyme denaturation, provided better reusability for the microbial biosensor after regeneration using 0.1 M K-phosphate buffer (pH 7.0) as the biosensor regeneration medium. The biosilica matrix reveals exceptional biocompatible support properties for biological molecules, with a benign immobilization method that retains enzyme activity. However, as the bio-doped microbial nanosilica was continuously exposed to the aqueous environment, the biosensor’s optical response began to decrease from the sixth reflectance measurement and onwards, due to leaching of the biorecognition elements from the sol-biogel composite during the continuous use of the biosensor. The biomimetic approach to microbial cell immobilization using the natural silaffin peptide with high affinity towards silica material has shown to provide outstanding repeatability performance, without covalent binding with biological receptor molecules, compared to some other biological immobilization methods, e.g., the entrapment and physisorption of uricase in/on poly(vinyl alcohol) *N*-methyl-4(4′-formylstyryl) pyridinium methosulfate acetal (PVA-SbQ) polymer matrix for the non-invasive detection of uric acid [42], and the crosslinking immobilization of penicillinase to ZnO though *N*-5-azido-2-nitrobenzoyloxysuccinimide crosslinker molecules for the electrochemical potentiometric determination of penicillin [43].

### 3.5. The Effect of Interferent on the Microbial Optosensor Response

In this study, the reflectance intensity of the microbial optosensor at 608 nm was evaluated in the presence of common nutrient ions (e.g., NO_3_^−^, NH_4_^−^, K^+^, Ca^2+^, Mg^2+^, Fe^2+^, and Fe^3+^) found in processed meats [44] using reflectometric optosensing of 20 mg L^−1^ nitrite in 0.1 M K-phosphate buffer at pH 7.0. The reflectance response of the microbial biosensor was measured at different concentration ratios between nitrite and potential interfering species at 1:1, 1:10, and 1:100, and the mean for the triplicate data was statistically compared with the mean for the reflectance intensity of the whole cell bacterial optosensing biomaterial without the presence of interferent by using the paired sample *t*-test technique. Table 1 tabulates the interference effect of potential interfering species at various concentrations on the optical nitrite biosensor response. 

No foreign ions showed significant interference effects to the optical response of the biosensor at low concentration levels (1:1), except the Fe^3+^ ion, due to its yellow-brown color in the solution that resulted in color interference to the optical response via absorption of higher light intensity transmitted through the feed fiber, attenuating the reflectance light response. Fe^2+^ ion is pale green, and it rendered a significant interference only at higher concentration levels (1:10). The color of the transition elements, or their derivative salts, is due to the excitation of unpaired electrons from the *d*-orbital of a lower energy state to the *d*-orbital of a higher energy state. Ca^2+^ and K^+^ ions are essential nutrients in animal nutrition. Both calcium and potassium contents can influence meat tenderness. Calcium is especially needed by the proteolytic system of calpain-calpastatin, the major factors in the postmortem tenderization of skeletal muscles [45]. The content of nitrate (NO_3_^−^) in processed meat products on the market is usually low, i.e., from 19–55 mg L^−1^ [13]. Ammonium (NH_4_^+^), Ca^2+^, and magnesium (Mg^2+^) ions caused an indicative interference effect on the optical nitrite biosensor response between nitrite and the interferent at a concentration ratio of 1:100. However, highly processed foods often tend to have a lower Mg^2+^ content, and the maximum allowable levels of ammonium salts that can be added to the processed foods, which do not pose risks to human health, is between 0.001% and 3.2% [46]. The Ca^2+^ ion interfered with the microbial optosensor response at high concentrations, as it formed a white precipitation of calcium hydroxide [Ca(OH)_2_] [47] and deviated the biosensor reflectance intensity at 608 nm. 

### 3.6. Validation of Microbial Optosensor with Standard Griess Method for Nitrite Quantitation

The use of nitrite in the production of dry-cured hams and dry sausages is beneficial for its excellent antioxidant and antimicrobial traits, while it also contributes to the cured meat flavor [48]. However, the addition of nitrite to processed meats is highly regulated because when consumed in excess, it is toxic to humans [14,49]. To allow in situ monitoring of nitrite, we attempted to demonstrate accurate nitrite estimation in processed meat products, such as nugget, sausage, and canned meats using the proposed nitrite optosensor based on biomimetic R5-fusion microbial nanosilica and validated using the standard UV-vis spectrophotometric Griess method. Table 2 tabulates the nitrite concentration results determined by both the UV-vis spectrophotometric Griess method and the reflectance microbial optosensor in processed meat sample matrices. The microbial biomimetic nanosilica showed 1.47 ± 0.56 mg L^−1^, 2.22 ± 0.45 mg L^−1^, and 3.71 ± 0.34 mg L^−1^ of nitrite concentrations present in the as-prepared nugget, sausage and canned meat liquid food samples, respectively, based on the reflectance intensity obtained at 608 nm. The Griess test involved two chemical reactions for the assay of nitrite. The sulfanilamide first reacted with the nitrite ion in the Griess diazotization reaction to form a diazonium salt, which then reacted with *N*-1-naphthylethylenediamine dihydrochloride (NED) in an azo coupling reaction to form a soluble reddish-pink azo compound, with a maximum absorption at the wavelength of 540 nm [50]. Upon treatment of the nitrite-containing meat liquid sample with the Griess reagent for 5 min, the UV-vis absorption of the resulting reddish-pink hue at 540 nm indicated that 1.63 ± 0.17 mg L^−1^, 2.50 ± 0.34 mg L^−1^, and 3.93 ± 0.34 mg L^−1^ nitrites were detected in the respective nugget, sausage, and canned meat samples. Both the optical biosensor and the UV-vis spectrophotometric Griess method exhibited the highest nitrite level in canned meat, followed by sausage and nugget. Statistical t test data analysis was applied to determine if the two methods showed a significant difference in the quantification of nitrite concentrations. The calculated *t* values obtained were then compared against the critical *t* value. For a significance level of 0.05 and 4 degrees of freedom, the critical value for the *t*-test (*t*_4_) is 2.776. Since the calculated *t* values of our test statistic were below the value of *t*_4_, this indicates the biosensor’s excellent capacity to provide an accurate mean value that is close to the mean value when the sample is measured by using a standard method for nitrite quantitation.

### 3.7. Comparison of the Developed Nitrite Biosensor with Several Recently Reported Optical Nitrite Sensors and Biosensors

Table 3 summarizes the comparison with regard to the analytical performance of the proposed reflectometric microbial optosensor with other recently reported sensors and biosensors for the optical detection of nitrite. The main advantages of the developed optical microbial biosensor based on the biomimetic silicification entrapment immobilization method are real-time rapid detection, satisfactory storage duration, and wide linear nitrite detection range for both field and laboratory analyses. The biosilicification synthetic route to silica nanoparticles revealed excellent robustness and superb biocompatibility that promoted diffusion and analyte uptake and access. 

Ho et al. [51] developed a portable liquid-crystal based sensor for nitrite detection with tetradecyl 4-aminobenzoate (14CBA)-doped 4-cyano-4′-pentylbiphenyl (5CB) nematic liquid crystal (LC) cast on a glass substrate. Nitrite detection was carried out from the basis of a bright-to-dark transition of the LC via diazotization reaction. However, the diazotization of alkylanilines reached equilibrium only after 60 min. A fluorimetric nitrite biosensor with polythienothiophene-fullerene thin film detectors based on integrated polymer photodetectors to detect light from fluorescence reactions with a diaminofluorescein probe [52], and fluorescence detection of nitrite by using neutral red (NR) molecules and gold nanoparticles (AuNPs), which were modified by per-6-mercapto-beta-cyclodextrins (SH-β-CDs), and changed in color in the presence of trace amount of nitrite [53], were able to detect nitrite to a submicromolar concentration. Nevertheless, these methods were confined to a narrow nitrite linear detection range at low levels. The incorporation of a copper(II) [Cu(II)] Schiff base complex into a transparent triacetylcellulose membrane produced from waste photographic film as an ionophore for a nitrite-selective optical sensor by absorption spectrophotometry [54], however, involved the use of toxic sodium hypochlorite, which is extremely reactive towards organic matter and is not friendly to the environment during the preparation of the nitrite sensor membrane.

## 4. Conclusions

A microbial optosensor based on biomimetic silaffin-fusion silica nanoparticles has been developed. The entrapment of *R. planticola* whole cell and non-protein NAD(P)H chemical compounds during biogenic nanosilica production created a favorable environment for the efficient diffusion of water-soluble analytes and reaction end-products within the biosilica matrix. The biomimetic silicification method has proven to be simple and efficient for the immobilization of biological molecules, compared to other physical and chemical immobilization techniques. The use of toxic chemicals and time-consuming methods were avoided by using a traditional sol-gel method for preparing silica materials, which could result in toxic residues in the final materials. Ruthenium(II)(bipy)_2_(phenanthroline-benzoylthiourea)-modified bio-doped microbial nanosilica conferred the visual detection of nitrite by virtue of it protonatable and deprotonatable phenanthroline-benzoylthiourea ligand. A validation study of the biomimetic nanosilica catalytic matrix revealed adequate feasibility of the microbial optosensor for use in processed meat samples for the detection of nitrite.

## Figures and Tables

**Figure 1 biosensors-12-00388-f001:**
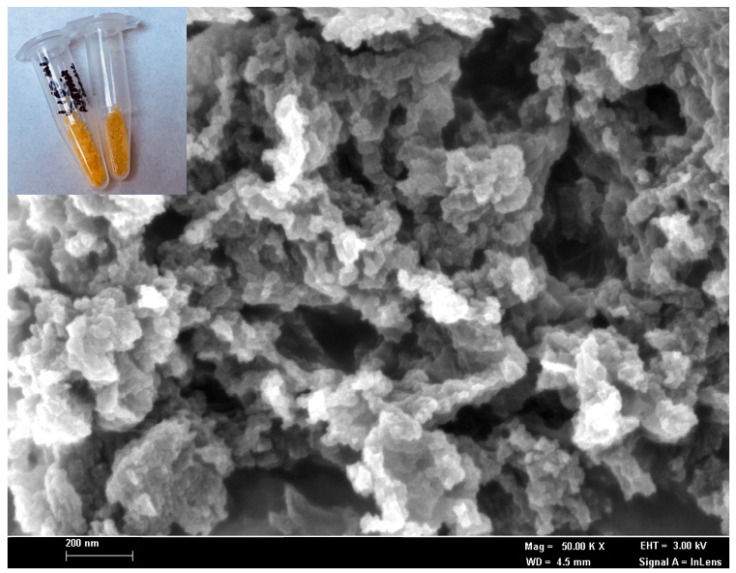
The FESEM micrograph, at a magnification of 50 k×, of the microbial R5-fusion nanosilica prepared by the biomimetic silicification process at ambient temperature and pressure. The inset shows that the photographic image of the corresponding R5-fusion microbial nanosilica appears to be orangish after immobilized with 0.1 mg mL^−1^ ruthenium optical pH probe at the surface via physical adsorption.

**Figure 2 biosensors-12-00388-f002:**
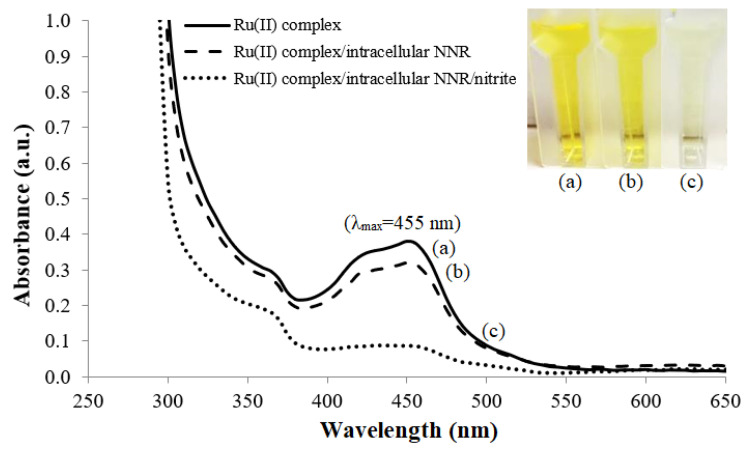
The UV-vis spectra of absorption peak responses at 455 nm for (**a**) 0.01 mg mL^−1^ Ru(II) complex solution, (**b**) free whole cell *R. planticola* and 0.048 mM NAD(P)H co-substrate in the 0.01 mg mL^−1^ Ru(II) complex solution before and (**c**) after reaction with nitrite in 0.1 M K-phosphate buffer at pH 7.4.

**Figure 3 biosensors-12-00388-f003:**
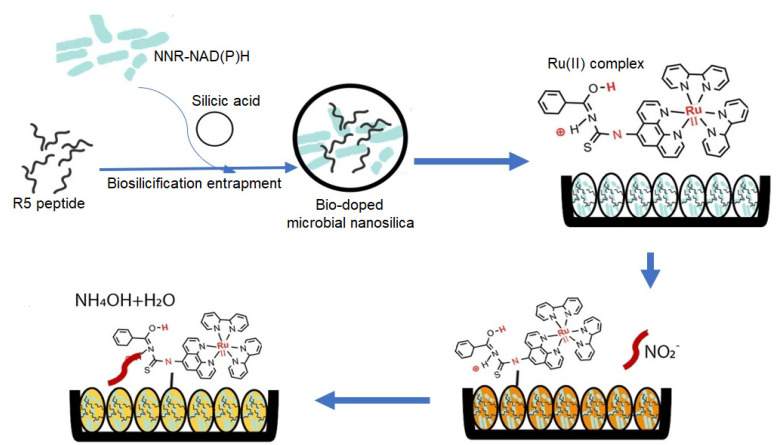
The schematic diagram represents the biomimetic silicification process involving R5 peptide from the silaffin protein of *C. fusiformis* to initiate the formation of silica nanoparticles with the co-silicification of the microbial intracellular NNR enzyme and cofactor NAD(P)H. Ruthenium complex, the pH optical label, is simply adsorbed to the surface of the R5-fusion nanosilica. The microbial optosensor changes from orange to yellow hue in the presence of nitrite, which can be monitored by the optical reflectometric method.

**Figure 4 biosensors-12-00388-f004:**
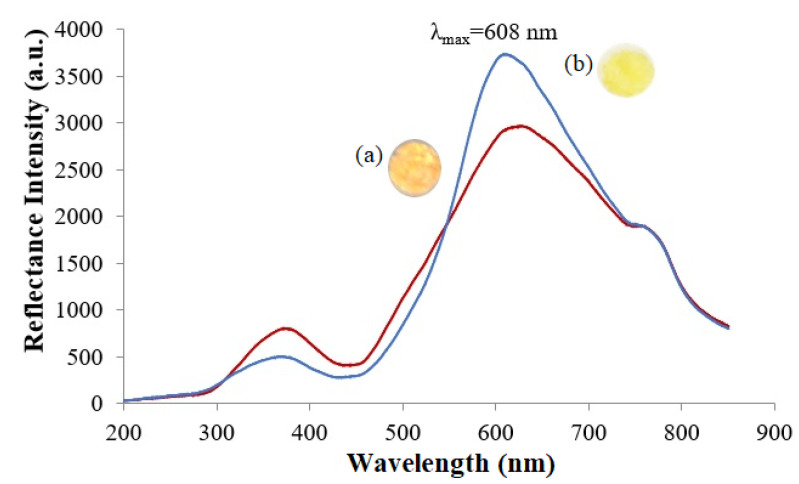
Reflectance spectra of the Ru(II) complex-modified biomimetic nanosilica support (**a**) before and (**b**) after reaction with 100 mg L^−1^ nitrite in 0.1 M K-phosphate buffer at pH 7.4. The bio-doped microbial nanosilica exhibited a maximum reflectance peak signal at 608 nm, and the reflectance peak signal significantly heightened due to the deprotonation of phen-BT ligands in the ruthenium complex, while the intracellular enzyme expression system catalyzed the enzymatic reduction of nitrite.

**Figure 5 biosensors-12-00388-f005:**
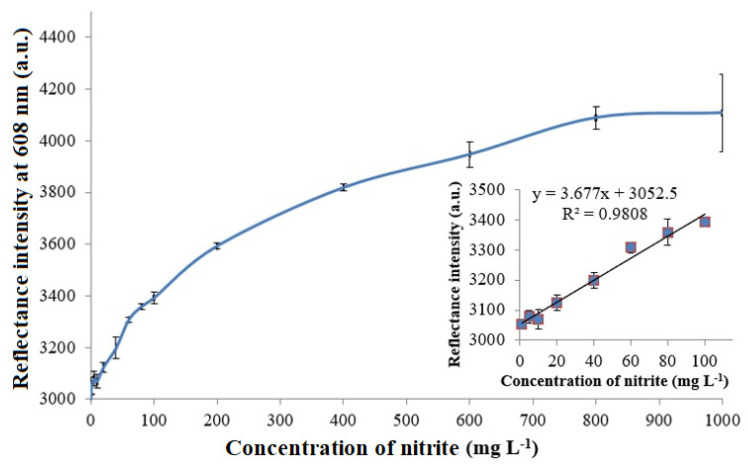
The nitrite concentration-dependent reflectometric microbial biosensor response at the nitrite concentration range of 1-1000 mg L^−1^ in 0.1 M K-phosphate buffer at a neutral pH. The inset shows the dynamic linear response range of the reflectance nitrite biosensor from 1–100 mg L^−1^ nitrite.

**Figure 6 biosensors-12-00388-f006:**
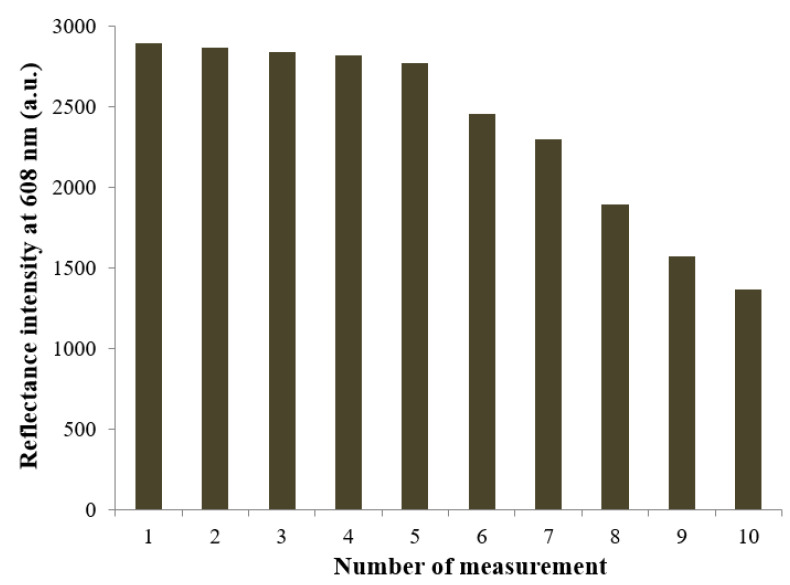
The repeatability performance of the reflectance microbial optosensor at 608 nm, which repeatedly reacted with 20 mg L^−1^ nitrite for 4 min after washing the bacterial cells biosensor with 0.1 M K-phosphate buffer at pH 7.0.

**Table 1 biosensors-12-00388-t001:** The reflectance intensity reading obtained from the developed microbial optosensor for the determination of 20 mg L^−1^ nitrite in the presence of common nutrient ions found in processed meats (*n* = 3).

Potential Interfering Species	Reflectance Intensity at Concentration Ratio between Nitrite and Potential Interfering Species, 608 nm (a.u.)			
	1:0	1:1	1:10	1:100
NO_3_^−^	2971.47 ± 0.79	3371.27 ± 0.71	3245.74 ± 0.46 ^a^	
NH_4_^+^	2971.47 ± 0.79	2901.41 ± 0.76	2967.94 ± 0.19	2842.22 ± 0.65 ^a^
K^+^	2971.47 ± 0.79	2968.80 ± 0.76	2761.83 ± 0.16 ^a^	
Ca^2+^	2971.47 ± 0.79	2858.51 ± 0.16	2828.42 ± 0.18	2749.58 ± 0.81 ^a^
Mg^2+^	2971.47 ± 0.79	3047.81 ± 0.19	2915.02 ± 0.19	2819.04 ± 0.87 ^a^
Fe^2+^	2971.47 ± 0.79	2872.67 ± 0.61	2716.51 ± 0.22 ^a^	
Fe^3+^	2971.47 ± 0.79	2873.07 ± 0.53 ^a^		

^a^*t* value exceeded *t* critical value at 95% confidence level.

**Table 2 biosensors-12-00388-t002:** Concentration of nitrite determined by both the UV-vis spectrophotometric Griess method and the reflectance microbial optosensor in processed meat products. The calculated *t* value is compared with the critical *t* value at 4 degrees of freedom and 95% confidence levels (*n* = 3).

Sample	Concentration of Nitrite (mg L^−1^)		*t* Value
	Griess Method	Microbial Optosensor	
Nugget	1.47 ± 0.56	1.63 ± 0.17	1.9956
Sausage	2.22 ± 0.45	2.50 ± 0.34	1.9993
Canned meat	3.71 ± 0.34	3.93 ± 0.34	1.9996

*t*_4_*=* 2.776 (*p* = 0.05).

**Table 3 biosensors-12-00388-t003:** Comparison of the developed microbial optosensor with several recently reported optical sensors and biosensors dedicated to nitrite detection with respect to linear detection range, detection limit, response time, and long-term stability.

Sensing Elements and Immobilization Matrix	Immobilization Technique	Detection Method	Linear Range (mg L^−1^)	Detection Limit (mg L^−1^)	Response Time (min)	Storage Stability (day)	Reference
*R. planticola*-NAD(P)H-Ru(II) complex-R5 fusion nanosilica	Biosilicification entrapment	Refelectance spectrophotometry	1.000–100.000	0.250	4	15	This work
Tetradecyl 4-aminobenzoate (14CBA)-doped 4-cyano-4′-pentylbiphenyl (5CB) liquid-crystal (LC)-glass slide	Casting of LC mixture onto glass substrate	Absorption spectrophotometry	1.725–690	1.725	60	-	Ho et al. [51]
Polythieno[3,4-*b*]thiophene/benzodithiophene (PTB7):PC70BM integrate organic photodetectors (OPDs)	Spin-coating	Fluorescence spectophotometry	0.076–1.932	0.025	14	-	Pires et al. [52]
Neutral red-SH-β-cyclodextrin (CD)@AuNPs	Chemical binding	Fluorescence spectophotometry	0.300–0.900	0.250	-	-	Du et al. [53]
Cu(II) Schiff base complex- triacetylcellulose membrane	Polymer entrapment	Absorption spectrophotometry	0.500–7.000	0.040	8–10	-	Habibzadeh et al. [54]

## Data Availability

Not applicable.

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
