# Peer review of "Bio-Doped Microbial Nanosilica as Optosensing Biomaterial for Visual Quantitation of Nitrite in Cured Meats"

_biosensors, 2022, doi:10.3390/bios12060388_

Round 1

Reviewer 1 Report

The article is well written. Also, interesting pieces of information were reported. However, spelling correction is required. The authors need to go through the article carefully before publication.  The following points need to be addressed before publication: (i) The authors need to justify why did they make reference to the Food Safety and Standards Authority of India. (ii) Better to use "FESEM micrograph". (iii) Figure and text missing the identification of absorption/reflectance spectra. (iv) Reference style is not uniform for all the references. The article may be considered for publication. 

Author Response

The article is well written. Also, interesting pieces of information were reported. However, spelling correction is required. The authors need to go through the article carefully before publication.  The following points need to be addressed before publication:

ANSWER- Spelling correction has been made throughout the manuscript for those marked in red.

(i) The authors need to justify why did they make reference to the Food Safety and Standards Authority of India.

ANSWER- Several cured meat samples, such as nugget, sausage and canned meat were prepared and pre-treated according to the manual of method for food analysis of Food Safety and Standards Authority of India [23], which provides clear-cut instructions as to how processed meat sample is prepared before analysis. Correction has been done in the abstract in page 4, line 195-196 of the revised manuscript.

(ii) Better to use "FESEM micrograph".

ANSWER- Correction has been done for the caption of Figure 1 in page 5, line 228 of the revised manuscript.

(iii) Figure and text missing the identification of absorption/reflectance spectra.

ANSWER- Corrections have been done for the caption of Figure 2 and Figure 4. The UV-vis and reflectance spectra are mentioned in the text in page 6, line 248-254 and page 8, line 298-303 respcetively of the revised mauscript.

(iv) Reference style is not uniform for all the references.

ANSWER- Correction has done in the in the reference section.

The article may be considered for publication. 

Reviewer 2 Report

The manuscript “Bio-doped microbial nanosilica as optosensing biomaterial for visual quantitation of nitrite in cured meats” (biosensors-1755146)contains an interesting topic about sensors. The experiment is well thought out and described. A very good introduction to the subject of research. The manuscript is suitable for publication in Biosensors.

I have a question for the authors about the alternative use of a simple ion-selective electrode test to analyze meat for the presence of nitrites.

1)      Ion-selective electrodes provide a wider measuring range than those presented in the paper (i.e. 1-10-5 M). Certainly, the constructing of new sensors is very important. Will a sensor working in the 1-100 mg range be convenient for rapid nitrite tests?

2)      I recommend that you complete the manuscript (in the Introduction section) about the use of an ion-selective electrode for nitrite analysis..

Author Response

The manuscript “Bio-doped microbial nanosilica as optosensing biomaterial for visual quantitation of nitrite in cured meats” (biosensors-1755146) contains an interesting topic about sensors. The experiment is well thought out and described. A very good introduction to the subject of research. The manuscript is suitable for publication in Biosensors. I have a question for the authors about the alternative use of a simple ion-selective electrode test to analyze meat for the presence of nitrites.

1) Ion-selective electrodes provide a wider measuring range than those presented in the paper (i.e. 1-10-5 M). Certainly, the constructing of new sensors is very important. Will a sensor working in the 1-100 mg range be convenient for rapid nitrite tests?

ANSWER- The dynamic linear response range covering the nitrite concentration between 1 mg L-1 and 100 mg L-1 of the microbial optosensor can be conveniently be used for rapid nitrite test as the immobilized ruthenium bipyridine complex pH optical label at the biomimetic nanosilica surface enabled visual detection of nitrite by simple color change method from orange to yellow of the bio-doped whole cell bacterial biosensor in 4 min.

2) I recommend that you complete the manuscript (in the Introduction section) about the use of an ion-selective electrode for nitrite analysis.

ANSWER- The ion-selective electrodes (ISEs) are part of a group of relatively simple and inexpensive analytical tools which are commonly referred to as sensors. The nitrite ISE electrode is intended for measuring nitrate ion concentrations and activities in aqueous solutions. Its preferred applications are to be found in the field of water, fertilizers, soil, meat, baby food analysis, etc. Commercially available ISEs for nitrite analysis that provide wider measuring range are such as Thomas brand nitrite ISE (0.5-460 mg L-1) [21]; NTsensors CNT_ISE (0.5-1000 mg L-1) [22], TRUEscience ISE (0.5-460 mg L-1) [23]; Orion nitrite electrode (0.5-200 mg L-1) [24]; and CLEAN instruments nitrite ISE sensor (0.2-220 mg L-1) [25]. These simple ISE tests can be used as the alternatives to analyze meat for the presence of nitrite. Correction has been done in the introduction section in page 3, line 108-117 of the revised manuscript.

References:

[21] Thomas Scientific, Nitrite ion selective electrode, 2022. Available online: https://www.thomassci.com/scientific-supplies/Nitrite-Ion-Selective-Electrode. (accessed on 30 May 2022).

[22] NTsensors, Nitrite ion selective electrode (ISE), 2022. Available online: https://www.ntsensors.com/parameters/nitrite-ion-selective-electrode/. (accessed on 30 May 2022)

[23] True Science, Replacement TRUEscience ion selective electrode for nitrite NO2–S7 no lead, 2022. Available online: https://www.truescience.co.uk/product/replacement-truescience-ion-selective-electrode-for-nitrite-no2-s7-no-lead/. (accessed on 30 May 2022)

[24] Thermo Electron Corporation, Orion nitrite electrode instruction electrode, 2022. Available online:  https://www.fondriest.com/pdf/thermo_nitrite_manual.pdf. (accessed on 30 May 2022)

[25] Clean Instruments-strider tech, A nitrite ion selective electrode sensor, 2022. Available online:  http://www.cleaninst.com/ions-nitrite.htm. (accessed on 30 May 2022)

Reviewer 3 Report

In this work, the authors reported on the nitrite biosensors prepared via a biomimetic approach. Briefly, the whole bacterial cells as well as the non-protein compounds were embedded into a biogenic silica matrix derived from the silaffin (R5) protein. For optical sensing, the authors incorporated a pH-sensitive Ru(II) complex probe whose color changes from orange to yellow in the presence of the analyte. The prepared biosensor has the advantage of being easy to fabricate and operate, having satisfactory analytical parameters, and being well suited for the detection of nitrite in cured meat samples. In addition, the sensor was fabricated without the usage of hazardous chemicals, which makes the whole process environmentally friendly and retains the enzyme activity as in the untrapped bacteria. I find the manuscript well written and easy to read. My comments are as follows:

Strengths:

1)      Fabrication of the biosensor under mild conditions; entrapment of R. planticola bacterial cells in a biocompatible biosilica matrix results in enzymatic activity over prolonged time.

2)      Wide linear detection range of the nitrite biosensor.

3)      Applicability in real samples.

Weaknesses:

1)      Lines 64, 79, 111, 201/202, 219, 223 – name of bacterial genus and species need to be written in italic.

2)      In lines 111 and 131, the definition TMOS is duplicated.

3)      In lines 134-136 it is stated that the model Gram-negative bacteria were isolated from bird nests. However, there is no mention of how the bacterial culture was stored prior to preparation of the R. planticola/NAD(P)H suspension for optosensor fabrication.

4)      Lines 203-208: Please explain, or provide additional citations, on how the size and shape of the prepared silica nanoparticles affect: a) the amount of Ru(II) probe physically adsorbed onto the surface of the R5-fused nanosilica particles and b) the biosensor analytical performance?

5)      Enzyme abbreviation for NNR enzymes needs to be explained in line 224 instead of line 247.

6)      Lines 325-327: Can you provide a KM value for a nitrite biosensor whose response follows Michaelis-Menten kinetics?

Author Response

In this work, the authors reported on the nitrite biosensors prepared via a biomimetic approach. Briefly, the whole bacterial cells as well as the non-protein compounds were embedded into a biogenic silica matrix derived from the silaffin (R5) protein. For optical sensing, the authors incorporated a pH-sensitive Ru(II) complex probe whose color changes from orange to yellow in the presence of the analyte. The prepared biosensor has the advantage of being easy to fabricate and operate, having satisfactory analytical parameters, and being well suited for the detection of nitrite in cured meat samples. In addition, the sensor was fabricated without the usage of hazardous chemicals, which makes the whole process environmentally friendly and retains the enzyme activity as in the untrapped bacteria. I find the manuscript well written and easy to read. My comments are as follows:

Strengths:

1) Fabrication of the biosensor under mild conditions; entrapment of R. planticola bacterial cells in a biocompatible biosilica matrix results in enzymatic activity over prolonged time.

2) Wide linear detection range of the nitrite biosensor.

3) Applicability in real samples.

Weaknesses:

1) Lines 64, 79, 111, 201/202, 219, 223 – name of bacterial genus and species need to be written in italic.

ANSWER- Correction to the name of bacterial genus and species has been done accordingly in the revised manuscript at line 66, 81, 124/125, 149,  219/220, 248/249 and 290.

2) In lines 111 and 131, the definition TMOS is duplicated.

ANSWER- Correction has been done in page 3, line 144 of the revised manuscript.

3) In lines 134-136 it is stated that the model Gram-negative bacteria were isolated from bird nests. However, there is no mention of how the bacterial culture was stored prior to preparation of the R. planticola/NAD(P)H suspension for optosensor fabrication.

ANSWER- R. planticola bacteria cells, which have been isolated from local edible bird’s nests from was stored at -80 °C. It was allowed to thaw on the ice prior to mixing with 2 mM NAD(P)H at a volume ratio of 3:1 for the fabrication of microbial optosensor. Correction has been done in page 3, line 149-150 and page 4, line 151-152 of the revised manuscript.

4) Lines 203-208: Please explain, or provide additional citations, on how the size and shape of the prepared silica nanoparticles affect: a) the amount of Ru(II) probe physically adsorbed onto the surface of the R5-fused nanosilica particles and b) the biosensor analytical performance?

ANSWER- As the R5-fusion biosilica matrix was synthesized by using TMOS as a silica source and that the biosilica matrix composed of four-dimensional silicon dioxide units with high porosity. Thereby, it is postulated that the ruthenium pH probe molecules were physically bound on the biomimetic nanosilica via hydrogen bonding since silica-based cell matrix is a suitable adsorbent for water, alcohol, phenols, amines and so on through hydrogen bonds at high loading capacity [31,32]. The intense orangish biosilica of R5 would allow a noticeable color change of the microbial optosensor ascribed to deprotonation of phen-BT ligands in the ruthenium complex, whilst intracellular enzyme expression system catalyzed enzymatic reduction of nitrite, and resulting in the yellowish substrate, which can be visually observed. Correction has been done in page 6, line 234-243 of the revised manuscript.

References:

[31] Ghaedi, M. Adsorption: fundamental processes and applications. Interface Sci. Technol. 2021, 3, 2-713. https://www.sciencedirect.com/topics/materials-science/silica-gel.

[32] Benitez-Medina, G.E.; Flores, R.; Vargas, L.; Cuenu, F.; Shama, P.; Castro, M.; Ramirez, A. Hybrid material by anchoring a ruthenium(II) imine complex to SiO2: preparation, characterization and DFT studies. RSC Adv. 2021, 11, 6221-6233. doi: 10.1039/D0RA09282G.

5) Enzyme abbreviation for NNR enzymes needs to be explained in line 224 instead of line 247.

ANSWER- Corrections have been in page 6, line 252 and page 7, line 277 of the revised manuscript.

6) Lines 325-327: Can you provide a KM value for a nitrite biosensor whose response follows Michaelis-Menten kinetics?

ANSWER- The apparent Michaelis-Menten constant (Km) of the nitrite biosensor estimated from the Lineweaver-Burk plot was 1.667 mM. Corrections have been in page 10, line 360-362 of the revised manuscript.
